# SelKD: Selective Knowledge Distillation via Optimal Transport Perspective

**Liangliang Shi**[1†], **Zhengyan Shi**[1†], **Junchi Yan**[12∗]
[1]Sch. of Computer Science & Sch. of Artificial Intelligence, Shanghai Jiao Tong University
[2]Shanghai Artificial Intelligence Laboratory
[†] Equal Contributions    [∗] Correspondence Author
`{shiliangliang,macho2021,yanjunchi}@sjtu.edu.cn`

## Abstract

Knowledge Distillation (KD) has been a popular paradigm for training a (smaller) student model from its teacher model. However, little research has been done on the practical scenario where only a subset of the teacher's knowledge needs to be distilled, which we term selective KD (SelKD). This demand is especially pronounced in the era of foundation models, where the teacher model can be significantly larger than the student model. To address this issue, we propose to rethink the knowledge distillation problem from the perspective of Inverse Optimal Transport (IOT). Previous Bayesian frameworks mapped each sample to the probabilities of corresponding labels in an end-to-end manner, which fixed the number of classification categories and hindered effective partial knowledge transfer. In contrast, IOT calculates from the standpoint of transportation or matching, allowing for the flexible selection of samples and their quantities for matching. Traditional logit-based KD can be viewed as a special case within the IOT framework. Building on this IOT foundation, we formalize this setting in the context of classification, where only selected categories from the teacher's category space are required to be recognized by the student in the context of closed-set recognition, which we call closed-set SelKD, enhancing the student's performance on specific subtasks. Furthermore, we extend the closed-set SelKD, introducing an open-set version of SelKD, where the student model is required to provide a "not selected" response for categories outside its assigned task. Experimental results on standard benchmarks demonstrate the superiority of our approach. The source code is available at: https://github.com/machoshi/SelKD.

## 1 Introduction

Knowledge Distillation (KD) (Hinton et al., 2015) has been a popular paradigm to transfer the knowledge from large models (teachers) to small ones (students), which has been widely used in different fields from visual recognition (Kong et al., 2019), speech recognition (Shen et al., 2020), natural language processing (Jiao et al., 2019), to recommendation systems (Pan et al., 2019). Many approaches have been proposed including matching the intermediate features (Romero et al., 2014), learning the relationships (Lee et al., 2018) and adopting the multiple teachers (Liu et al., 2020).

Existing KD methods typically transfer the entire knowledge from one (Sun et al., 2024) or multiple (Yuan et al., 2021) teacher models to a student model. However, in many real-world applications, it is often preferable for the student model to learn only a subset of the teacher's knowledge. This scenario becomes particularly relevant when the teacher is a large foundation model, while the student model is deployed in resource-constrained environments such as edge computing. Despite its practical significance, this setting has received little attention in prior work.

To address this gap, we formalize the described setting as selective knowledge distillation (SelKD) within the context of classification. Unlike traditional KD, SelKD requires the specification of

[∗]This work was in part supported by NSFC (62222607) and Shanghai Municipal Science and Technology Major Project under Grant 2021SHZDZX0102.

categories (i.e., subsets of knowledge)[1] as a side input, allowing the student model to focus exclusively on learning this selected knowledge. This targeted approach makes SelKD particularly applicable to real-world scenarios, where efficiency and task-specific learning are crucial.

SelKD has practical applications in real-world scenarios. Typically, we tend to implement more complex functionalities in relatively larger networks, which often run on high-performance servers. However, for smaller devices such as smartphones and tablets with limited computational power, it is often unnecessary to replicate the full functionality of models running on large servers. Instead, they may only need to perform specific tasks that are tailored to the device's capabilities. To improve the model performance on different devices, traditional knowledge distillation methods require training multiple teacher models for specific tasks to ensure consistency between teachers and students. However, the advantage of the SelKD framework lies in the fact that we only need to train a single strong teacher classifier capable of recognizing a wide range of categories. This teacher model can then be utilized to selectively transfer the relevant knowledge to different students with their respective subtasks. As a result, there is no need to retrain a teacher for each specific task, leading to reduced computational costs and simplified training process.

In this paper, we adopt the inverse optimal transport (IOT) perspective to address the classification problem. We define labels as a set of features, such as one-hot vectors or features extracted by a text encoder. Our goal is to establish a matching or transportation (i.e., coupling) between the features of images and texts. In this context, the learning process can be seen as the inverse of Entropic Optimal Transport, while the testing inference can be viewed as the optimization of Optimal Transport. From this perspective, we can naturally define the student categories as a subset of the categories of the teacher, enabling the knowledge transfer in the SelKD setting.

We propose two distinct settings within our SelKD framework. The first, referred to as (closed-set) SelKD, focuses on the teacher transferring knowledge related only to a specific subtask. The student model is trained exclusively on the data relevant to the assigned subtask and is not required to recognize categories beyond this scope. The second setting introduces open-set SelKD, which extends the framework to handle the recognition of unselected classes. Specifically, in resource-constrained devices, if a sample falls outside the subtask's recognition domain, open-set SelKD enables the student to provide a "not selected" or "reject due to unknown" response. To tackle this challenge, we employ a modified inverse optimal transport approach that relaxes the Softmax constraint, allowing the row-sum to be less than 1. The contributions of this paper can be summarized as follows:

1) We revisit the Knowledge Distillation (KD) problem through the lens of Inverse Optimal Transport (IOT), reformulating the vanilla KD problem as a bi-level optimization task. In the inner optimization, the goal is to learn the coupling (i.e., the matching probability) of the student model, which is then supervised by both the ground truth and the teacher's coupling to update the model parameters.

2) Building on this IOT-based formulation, we introduce Selective Knowledge Distillation (SelKD), where the student model is trained to learn only specific subtasks from the teacher model. Additionally, by adjusting the constraints of the original closed-set SelKD for open-set setting, we propose an open-set version of SelKD. The open-set SelKD requires the model to recognize "not selected" knowledge, allowing for a more flexible and robust response to unassigned tasks.

3) Our proposed method demonstrates superior performance compared to state-of-the-art techniques in both closed-set and open-set SelKD tasks, achieving notable improvements in accuracy and robustness. This highlights the effectiveness of the IOT-based KD approach and its potential for diverse real-world applications.

## 2 RELATED WORKS

### 2.1 KNOWLEDGE DISTILLATION

**Logit-based KD.** The idea of training smaller, cheaper models (students) to mimic larger ones (teachers) can be dated back to (Bucila et al., 2006) and it has been applied to neural networks among various tasks including classification (Hinton et al., 2015), speech recognition (Shen et al.,

---

[1]In the context of classification, including open-set settings, we use the terms "categories," "knowledge," and "subtasks" interchangeably to refer to the designated portions for selective knowledge distillation.

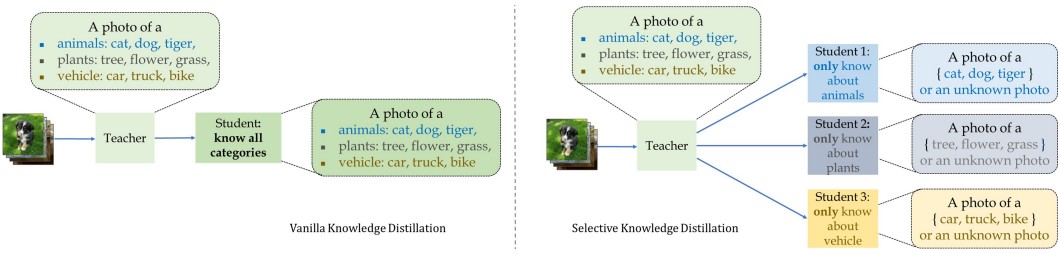

Figure 1: Illustrative comparison between vanilla knowledge distillation and our selective knowledge distillation with regard to classification tasks. In the vanilla KD framework, the student model learns knowledge from all categories in the teacher model. In contrast, within the selective KD framework, each student independently learns only a subset of categories from the teacher model, with the collective knowledge acquired by the students encompassing the entirety of the teacher's knowledge.

2020), natural language processing (Jiao et al., 2019), Large-scale language-image pretraining (Wu et al., 2023) etc. From a broader perspective, KD can be categorized into three types based on how the students learn knowledge from the teachers: logit-based, feature-based and relation-based KD. In particular, the logit-based KD methods distill the knowledge by aligning the logits between the teacher and student, which can be formulated as a loss as follows (Hinton et al., 2015):

$$\mathcal{L}_{\text{logit}} = CE(y \mid \sigma(f_s(\mathbf{x}); \tau)) + \lambda \cdot KL(\sigma(f_t(\mathbf{x}); \tau) \parallel \sigma(f_s(\mathbf{x}); \tau)), \tag{1}$$

where $f_s(\cdot)$ and $f_t(\cdot)$ are the sample encoders of the student and teacher models, respectively, and $y$ denotes the ground truth (i.e., label) of the sample. $\sigma(\cdot)$ is the softmax function mapping the logits to the category probabilities and $\tau$ is the temperature to control the smoothness of predictive distribution. $CE$ and $KL$ denotes the cross entropy loss and KL divergence, respectively. The parameter $\lambda$ controls the weight between the two items. The concept of logit-based knowledge distillation is straightforward and becomes particularly intuitive when viewed as a process of knowledge transfer. From another perspective, the effectiveness of soft targets can be compared to techniques such as label smoothing (Kim & Kim, 2017) or regularization methods (Müller et al., 2019; Ding et al., 2019). However, traditional logit-based distillation typically relies on the output of the final layer, like soft targets, which overlooks intermediate-level supervision from the teacher model—an essential component for effective representation learning in very deep neural networks (Romero et al., 2014). Additionally, since soft logits reflect class probability distributions, logit-based distillation is inherently limited to supervised learning scenarios.

**Feature-based KD.** In addition to logit-based KD methods, feature-based KD methods (Yang et al., 2023) primarily focus on aligning the intermediate features between the teacher and student models. This alignment can be expressed as:

$$\mathcal{L}_{\text{feature}} = CE(y \mid \sigma(\mathbf{F}^s; \tau)) + \lambda \cdot \mathcal{D}_{\text{feature}}(T_t(\mathbf{F}^t) \parallel T_s(\mathbf{F}^s)), \tag{2}$$

where $\mathbf{F}^t$ and $\mathbf{F}^s$ represent the intermediate features from the teacher and student models, respectively. $T_t$ and $T_s$ are feature transformation mappings for the teacher and student models, used to align the dimensions of $\mathbf{F}^t$ and $\mathbf{F}^s$. The term $\mathcal{D}_{\text{feature}}$ measures the divergence to quantify the feature difference between the two models, and the parameter $\lambda$ controls the weight between the two items.

**Self KD.** In self KD, the student model itself plays the role of the teacher. Inspired by the analysis of label smoothing regularization, a teacher–free KD method is proposed in (Yuan et al., 2019), whose core idea involves the model generating soft labels from its own knowledge and using these labels for training. (Yang et al., 2022) suggests integrating self KD with image mixture and aggregating multi-stage features to generate soft labels. In the paper (Li, 2022), channel features and layer features are utilized to transfer knowledge without the need for an additional model. To conclude, the main advantage of self KD is that it allows training a student model with a smaller teacher model size, while achieving performance comparable to the student model trained using a larger teacher model. Compared to SelKD, the teacher model in self KD shares the same task to the students.

## 2.2 Optimal Transport and Inverse Optimal Transport

As originally introduced by (Kantorovich, 1942), Optimal Transport (OT) (Raghvendra et al., 2024) involves solving a linear program (Lahn et al., 2023; Phatak et al., 2023), and has become a widely used tool across various research fields, including visual matching (Wang et al., 2013), long-tailed learning (Shi et al., 2024b;c), time series analysis (Zhang et al., 2020; Shi et al., 2020), multi-modal learning (Shi et al., 2024a; Wang et al., 2024), and more. Specifically, given the cost matrix $\mathbf{C}$ and two histograms (i.e., probability vectors) $\mathbf{a} \in \mathbb{R}^n, \mathbf{b} \in \mathbb{R}^m$ , Kantorovich's OT involves solving the coupling $\mathbf{P}$ (i.e., the joint probability matrix) by

$$\min_{\mathbf{P} \in U(\mathbf{a},\mathbf{b})} \langle \mathbf{C}, \mathbf{P} \rangle = \sum_{i=1}^{n} \sum_{j=1}^{m} \mathbf{C}_{ij} \mathbf{P}_{ij}, \tag{3}$$

where $U(\mathbf{a}, \mathbf{b})$ is the set of the couplings:

$$U(\mathbf{a}, \mathbf{b}) = \{\mathbf{P} \in \mathbb{R}_+^{n \times m} \mid \mathbf{P}\mathbf{1}_m = \mathbf{a}, \mathbf{P}^\top \mathbf{1}_n = \mathbf{b}\}. \tag{4}$$

which is bounded and defined by $n + m$ equality constraints.

A lot of methods (Bertsimas & Tsitsiklis, 1997; Benamou & Brenier, 2000) are proposed to solve the Kantorovitch OT problem and relaxing with the entropic regularization (Wilson, 1969) is one of the simple but efficient methods, whose objective reads:

$$\min_{\mathbf{P} \in U(\mathbf{a},\mathbf{b})} \langle \mathbf{C}, \mathbf{P} \rangle - \epsilon H(\mathbf{P}), \tag{5}$$

where $\epsilon > 0$ is the coefficient for entropic regularization $H(\mathbf{P})$ and the $H(\mathbf{P})$ can be specified as

$$H(\mathbf{P}) = -\sum_{i,j} \mathbf{P}_{ij} (\log(\mathbf{P}_{ij} - 1)). \tag{6}$$

The objective in Eq. 5 is an $\epsilon$-strongly convex function, and thus the optimization has a unique solution, which can be solved with iterative methods (e.g. the Sinkorn method (Sinkhorn, 1967)). If we use this entropic regularized OT to solve the matching problem, the hard matching problem may convert to soft matching.

Inverse Optimal Transport (IOT) has been explored in several studies (Dupuy et al., 2016; Li et al., 2019; Stuart & Wolfram, 2020), aiming to infer the unknown cost matrix $\mathbf{C}$ that generates the observed coupling. The work by (Stuart & Wolfram, 2020) presents a systematic approach for inferring these unknown costs, while (Chiu et al., 2022) develops the mathematical theory underpinning IOT. In addition, (Shi et al., 2023) demonstrates a brand new series of contrastive losses with set matching based on IOT. The IOT problem can be formulated as a bi-level optimization problem:

$$\min_{\theta} KL(\tilde{\mathbf{P}} \mid \mathbf{P}^\theta) \quad \text{where} \quad \mathbf{P}^\theta = \arg \min_{\mathbf{P} \in U(\mathbf{a},\mathbf{b})} \langle \mathbf{C}^\theta, \mathbf{P} \rangle - \epsilon H(\mathbf{P}). \tag{7}$$

where $\tilde{\mathbf{P}}$ is the ground truth for supervision. IOT facilitates the capture of fine-grained relationships between sample features, thereby enhancing the transfer of structured knowledge from the teacher model to the student model during the knowledge distillation process.

## 3 Methodology and Formulations

In contrast to vanilla KD where the student learns all the information from the teacher, we propose the setting of Selective KD (SelKD) that transfers only selective knowledge to the student. Without loss of generality, in this paper we view the classification task with optimal transport, in which labels are defined as a set of features (e.g. one-hot vectors or features extracted by a text encoder) and images are also represented with features extracted by an image encoder. Our goal is to establish a match or transportation between the features of images and texts with the formulation of OT. In this case, variants of optimal transport with specific properties could be introduced to solve the KD problem.

### 3.1 Optimal Transport Formulated (Logit-based) Knowledge Distillation

**Teacher Training via IOT perspective.** Given the batch data $\{(\mathbf{x}_i, \mathbf{y}_i)_{i=1}^N\}$, where $\mathbf{y}_i$ is the one-hot vector corresponding to sample $\mathbf{x}_i$ and $N$ is the batch size, the features of the samples and labels can

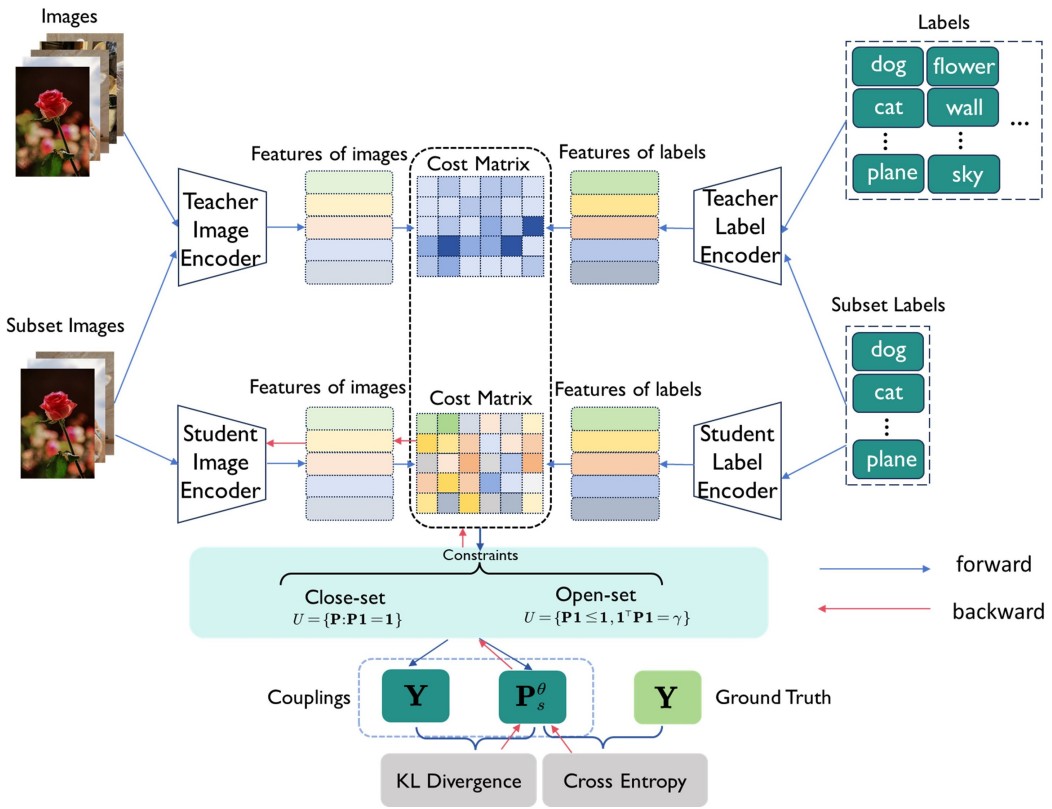

Figure 2: The overview of our approach for both closed-set and open-set SelKD tasks. We first compute cost matrices with features extracted from image samples and labels with encoders (CNN) respectively. The regularized OT is used to analyze and estimate the coupling of the student, which is supervised with ground truth and the coupling of the teacher for representation learning.

be represented by two sets: $\{f_\theta(\mathbf{x}_i)\}_{i=1}^N$, $\{g_\theta(\mathbf{y}_j)\}_{j=1}^M$. The label matrix is denoted as $\mathbf{Y} = \{\mathbf{y}_j\}_{j=1}^M$. The training process of the teacher model can then be reformulated as a bi-level optimization:

$$\min_\theta KL(\mathbf{Y} \mid \mathbf{P}^\theta) \quad \text{where} \quad \mathbf{P}^\theta = \arg\min_{\mathbf{P1}=1} \langle \mathbf{C}^\theta, \mathbf{P} \rangle - \epsilon H(\mathbf{P}). \tag{8}$$

Here the cost matrix $\mathbf{C}^\theta \in \mathbb{R}_+^{n \times m}$ is designed with features $\{f_\theta(\mathbf{x}_i)\}_{i=1}^N$, $\{g_\theta(\mathbf{y}_j)\}_{j=1}^M$ with parameters $\theta$ from the networks $f_\theta$ and $g_\theta$. This bi-level optimization consists of an outer and an inner optimization. As proved in (Shi et al., 2023), the outer optimization can be regarded as minimization of the cross-entropy loss, while the inner optimization, resembling an optimal transport (OT) problem, operates like a Softmax function, with the constraint $\mathbf{P1} = \mathbf{1}$ ensuring that the probabilities across each category sum to 1. A detailed explanation can be found in Appendix B. These constraints can be interpreted as simplified versions of the row and column sum constraints $U(\mathbf{a}, \mathbf{b})$ in OT. From this perspective, the teacher's learning process can be viewed as a bi-level optimization within the framework of entropic regularized OT. Compared to the Bayesian approach, this method offers greater flexibility in selecting desired categories, making it well-suited for implementing SelKD framework.

**(Logit-based) KD via IOT perspective.** Inspired by the Inverse Optimal Transport (IOT) framework, as described in Eq. 7, we propose a novel approach to integrate IOT constraints into the knowledge distillation (KD) process. Using logit-based KD as an example, we reformulate the knowledge distillation process as a bi-level optimization problem:

$$\min_\theta CE(\mathbf{Y} \mid \mathbf{P}_s^\theta) + \lambda \cdot KL(\mathbf{P}_t \parallel \mathbf{P}_s^\theta),$$

$$\text{where} \quad \mathbf{P}_t = \arg\min_{\mathbf{P1}=1} \langle \mathbf{C}_t, \mathbf{P} \rangle - \epsilon H(\mathbf{P}) \quad \mathbf{P}_s^\theta = \arg\min_{\mathbf{P1}=1} \langle \mathbf{C}_s^\theta, \mathbf{P} \rangle - \epsilon H(\mathbf{P}). \tag{9}$$

Here $H(\mathbf{P})$ is the entropic regularization as defined in Eq. 6, and $\mathbf{Y}$ denotes the label matrix. In this optimization, the cost matrix $\mathbf{C}_t \in \mathbb{R}_+^{n \times m}$ is designed with features from the pre-trained teacher

networks $f_t$ and $g_t$, while $\mathbf{C}_s^\theta \in \mathbb{R}_+^{n \times m}$ is computed with features $\{f_{s,\theta}(\mathbf{x}_i)\}_{i=1}^N$, $\{g_{s,\theta}(\mathbf{y}_j)\}_{j=1}^M$ with parameters $\theta$ from the networks $f_{s,\theta}$ and $g_{s,\theta}$. Different from previous works setting $U = \{\mathbf{P} : \mathbf{P1} = \mathbf{1}\}$, we think the constraint $U$ can be designed according to the specific circumstances of the problem, especially in the case of open-set tasks. We will discuss it in detail in the next subsection. In the outer minimization, the coupling $\mathbf{P}_s^\theta$ is calculated by OT's inner minimization and $\mathbf{Y}$ is the ground truth for supervision.

The aim of outer minimization is to supervise the student coupling with the ground truth and the teacher coupling, in order to learn the feature extractor (i.e. $f_{s,\theta}(\cdot)$ and $g_{s,\theta}(\cdot)$). Simultaneously, the inner minimization formulates the distillation problem as an entropic regularized optimal transport task. Our overarching goal is to derive the student coupling $\mathbf{P}_s^\theta$ that aligns with the teacher coupling $\mathbf{P}_t$, utilizing the respective cost matrices $\mathbf{C}_s^\theta$ and $\mathbf{C}_t$.

To summarize, this innovative perspective on logit-based knowledge distillation (KD) through the lens of Inverse Optimal Transport (IOT) reformulates the distillation process as a bi-level optimization problem. By incorporating IOT constraints, it allows for a more structured approach to aligning the teacher's and student's knowledge. In addition, the IOT-based view leads to the introduction of Selective Knowledge Distillation (SelKD), which focuses on targeted and efficient knowledge transfer and allows the student model to selectively learn relevant knowledge for specific subtasks.

## 3.2 SELECTIVE KNOWLEDGE DISTILLATION VIA INVERSE OPTIMAL TRANSPORT

In traditional knowledge distillation, the primary objective is to transfer the entire knowledge of the teacher model to the student model in a straightforward end-to-end manner. However, this approach can be inefficient and overly complex, especially in real-world applications where a student model may only need to perform a subset of tasks. Our proposed method, Selective Knowledge Distillation (SelKD), addresses this limitation by enabling the student model to learn only the relevant knowledge from the teacher model for specific subtasks, with evaluations focused solely on those areas during testing. Figure 1 illustates the difference between vanilla KD and our proposed SelKD.

Leveraging the perspective of Inverse Optimal Transport (IOT) in Eq. 9, we can formalize a general bi-level optimization framework for SelKD, expressed as follows:

$$
\min_\theta CE(\mathbf{Y} \mid \mathbf{P}_s^\theta) + \lambda \cdot KL(\mathbf{P}_t \parallel \mathbf{P}_s^\theta),
$$
$$
\text{where} \quad \mathbf{P}_t = \arg\min_U \langle \mathbf{C}_t, \mathbf{P} \rangle - \epsilon H(\mathbf{P}) \quad \mathbf{P}_s^\theta = \arg\min_U \langle \mathbf{C}_s^\theta, \mathbf{P} \rangle - \epsilon H(\mathbf{P}). \tag{10}
$$

In this formulation, $U$ represents the general constraints for the coupling. A key advantage of this approach is that it allows us to tailor the cost matrices $\mathbf{C}_s^\theta$ and $\mathbf{C}_t$ according to the specific categories we select for distillation. This flexibility enables a more nuanced and effective knowledge transfer, as the constraints can be adjusted based on the particularities of the tasks at hand.

In the following sections, we will detail how to specify the constraints $U$ and select feasible categories to formalize the optimization process, whether in closed-set or open-set scenarios.

### 3.2.1 (CLOSED-SET) SELECTIVE KNOWLEDGE DISTILLATION

We begin by analyzing Eq. 10 under the constraint $U = \{\mathbf{P} : \mathbf{P1} = \mathbf{1}\}$. In this context, we partition the training set based on the selected categories. Specifically, we define the complete category set as $\mathcal{C} = \{1, 2, \cdots, N\}$ where $N$ represents the total number of categories. We set the entire dataset $\mathcal{S} = \{(\mathbf{x}_i, \mathbf{y}_i)_{i=1}^M\}$ where $M$ is the size of dataset and $\mathbf{y}_i$ is the one-hot vector corresponding to sample $\mathbf{x}_i$. Without loss of generality, denoting the selected categories set $\mathcal{C}_{\text{closed-set}} = \{1, 2, \cdots, n\} \subset \mathcal{C}$ with $n < N$, we denote $\mathcal{S}_{\text{closed-set}} = \{(\mathbf{x}_i, \mathbf{y}_i) \mid \mathbf{y}_i \in \text{one-hot}(\mathcal{C}_{\text{closed-set}})\}$. Here one-hot$(\cdot)$ is a mapping function that converts all elements in the set to their one-hot representations.

For the batch data $\{(\mathbf{x}_i, \mathbf{y}_i)_{i=1}^m\} \subset \mathcal{S}_{\text{closed-set}}$ where $m$ is the batch size, the features of samples and labels can be represented by two sets $\{\{f_{s,\theta}(\mathbf{x}_i)\}_{i=1}^m, \{g_{s,\theta}(\mathbf{y}_j)\}_{j=1}^n\}$ and $\{\{f_t(\mathbf{x}_i)\}_{i=1}^m, \{g_t(\mathbf{y}_j)\}_{j=1}^n\}$ regarding student and teacher extracted features. Denoting $\mathcal{S}_{\text{image}} = \{(\mathbf{x}_i)_{i=1}^m\}$, without loss of generality, we set $\mathbf{C}_s$ and $\mathbf{C}_t$ as follows:

$$
(\mathbf{C}_s^\theta)_{ij} = -f_{s,\theta}(\mathbf{x}_i) \cdot g_{s,\theta}(\mathbf{y}_j) \quad (\mathbf{C}_t)_{ij} = -f_t(\mathbf{x}_i) \cdot g_t(\mathbf{y}_j) \quad \text{for} \quad i \in \mathcal{S}_{\text{image}}, j \in \mathcal{C}_{\text{closed-set}} \tag{11}
$$

For traditional classifiers, $f(\cdot)$ represents the image encoder and $g(\cdot)$ represents the label encoder (e.g., one-hot encoding). For multimodal classifiers based on CLIP (Radford et al., 2021), $f(\cdot)$ represents the image encoder while $g(\cdot)$ represents the text encoder. Our formulation generalizes this approach, making it more adaptable to a wider range of methods.

Figure 2 illustrates the pipeline of our method. Initially, we compute the cost matrices using features extracted from image samples and labels via encoders (CNN). These cost matrices are then incorporated into Eq. 11, forming the optimization formula for SelKD. To facilitate understanding, the cost matrices can be interpreted as the negative of similarity matrices (ignoring constant factors in the optimization problem).

### 3.2.2 OPEN-SET VERSION FOR SELECTIVE KNOWLEDGE DISTILLATION

We propose open-set SelKD, an extension of the SelKD framework mentioned above, designed to address the recognition of classes that were not included in the student's training subset. Specifically, in smaller or resource-constrained devices, SelKD tasks does not allow the student model to be trained on the full set of classes from the teacher model. Open-set SelKD requires student models to handle cases where an input sample falls outside the scope of the subtask that the student has been trained to recognize. In such situations, rather than attempting to force a classification, the student model is equipped to produce a "not selected" or "reject due to unknown" response, effectively acknowledging that the sample does not belong to any of the known classes. This ability to reject unknown inputs enhances the robustness and applicability of SelKD, especially in open-set or dynamic environments where new or unseen classes may emerge.

---

**Algorithm 1:** Computing the KL divergence between student and teacher couplings under Open-set SelKD

---

**Input** : features $\mathbf{F}_{\text{image},t}$ and $\mathbf{F}_{\text{image},s}$ extracted from image samples, $\mathbf{F}_{\text{label},t}$ and $\mathbf{F}_{\text{label},s}$ extracted from labels, (footnotes $s$ or $t$ represents student and teacher models, respectively), temperature $\tau$ and parameter $\gamma$

**Output :** Loss function $\mathcal{L}$

Initialize $\mathbf{C}_s = \mathbf{F}_{\text{image},s}\mathbf{F}_{\text{label},s}^\top$ and $\mathbf{C}_t = \mathbf{F}_{\text{image},t}\mathbf{F}_{\text{label},t}^\top$

Compute $\mathbf{P}_s^{(0)} = e^{-\mathbf{C}_s/\tau}$ and $\mathbf{P}_t^{(0)} = e^{-\mathbf{C}_t/\tau}$

**for** $l = 1, 2, \cdots, L$ **do**

$\quad \mathbf{P}_s^{(l)} = \dfrac{\mathbf{P}_s^{(l)}}{\max(\mathbf{P}_s^{(l)}\mathbf{1},\mathbf{1})}$

$\quad \mathbf{P}_s^{(l+1)} = \mathbf{P}_s^{(l)} \cdot \dfrac{\gamma}{\mathbf{1}^\top \mathbf{P}_s^{(l)}\mathbf{1}}$

$\quad \mathbf{P}_t^{(l)} = \dfrac{\mathbf{P}_t^{(l)}}{\max(\mathbf{P}_t^{(l)}\mathbf{1},\mathbf{1})}$

$\quad \mathbf{P}_t^{(l+1)} = \mathbf{P}_t^{(l)} \cdot \dfrac{\gamma}{\mathbf{1}^\top \mathbf{P}_t^{(l)}\mathbf{1}}$

**end**

**return** $\mathcal{L} = -\mathbf{1}^\top (\mathbf{P}_t \cdot \log(\mathbf{P}_s))\mathbf{1}$

---

To formulate open-set SelKD with an optimization based on the closed-set version, we first relax the constraints by setting $\mathbf{P1} = \mathbf{1}$ to a new one motivated by Partial Optimal Transport given as

$$U_{\text{M-POT}} = \{\mathbf{P1} \leq \mathbf{1}, \mathbf{1}^\top \mathbf{P1} = \gamma\} \tag{12}$$

where $\gamma$ is is the number of batch sample classified to the categories of the subtask.

Similarly, we define the complete category set as $\mathcal{C} = \{1, 2, \cdots, N\}$ where $N$ represents the total number of categories. We set the entire dataset $\mathcal{S} = \{(\mathbf{x}_i, \mathbf{y}_i)_{i=1}^M\}$ where $M$ is the size of dataset and $\mathbf{y}_i$ is the one-hot vector corresponding to sample $\mathbf{x}_i$. Without loss of generality, denoting the selected categories set $\mathcal{C}_{\text{closed-set}} = \{1, 2, \cdots, n\} \subset \mathcal{C}$ with $n < N$, we denote $\mathcal{C}_{\text{open-set}} = \mathcal{C}_{\text{closed-set}} \cup \{n + 1\}$ where $n + 1$ represents the union of "not selected" categories in the student model.

For the batch data $\{(\mathbf{x}_i, \mathbf{y}_i)_{i=1}^m\} \subset \mathcal{S}$, the features of samples and labels can be represented by two sets $\{\{f_{s,\theta}(\mathbf{x}_i)\}_{i=1}^m, \{g_{s,\theta}(\mathbf{y}_j)\}_{j=1}^{n+1}\}$ and $\{\{f_t(\mathbf{x}_i)\}_{i=1}^m, \{g_t(\mathbf{y}_j)\}_{j=1}^N\}$ with footnotes. Denoting $\mathcal{S}_{\text{image}} = \{(\mathbf{x}_i)_{i=1}^m\}$, without loss of generality, we set $\mathbf{C}_s$ and $\mathbf{C}_t$ as follows:

$$(\mathbf{C}_s^\theta)_{ij} = -f_{s,\theta}(\mathbf{x}_i) \cdot g_{s,\theta}(\mathbf{y}_j) \quad (\mathbf{C}_t)_{ij} = -f_t(\mathbf{x}_i) \cdot g_t(\mathbf{y}_j) \quad \text{for} \quad i \in \mathcal{S}_{\text{image}}, j \in \mathcal{C}_{\text{open-set}} \tag{13}$$

Table 1: Top-1 accuracy (%) on CIFAR-100 for closed-set SelKD. We compare the performance of vanilla knowledge distillation (KD), self-knowledge distillation (SKD) and our selective knowledge distillation (SelKD) in the closed-set SelKD classification tasks. All the comparisons are conducted based on feature-based distillation (FitNet Romero et al. (2014), FT Kim et al. (2018) and AT Zagoruyko & Komodakis (2016)).

| Student Networks | Methods | SubTask 0 | SubTask 1 | SubTask 2 | SubTask 3 | SubTask 4 | Mean of SubTasks | #Param of Teachers + Students |
|---|---|---|---|---|---|---|---|---|
| ResNet-8 | Without KD | 81.00 | 82.45 | 80.85 | 76.90 | 83.30 | 80.90 | 0 + 0.4M |
| | FitNet-KD | 81.50 | 83.55 | 82.20 | 78.50 | 85.65 | 82.28 | 117.7M + 0.4M |
| | FitNet-SKD | 81.70 | 83.25 | 81.55 | 77.75 | 85.05 | 81.86 | 0.4M + 0.4M |
| | FitNet-SelKD | **82.95** | **84.45** | **82.25** | **78.85** | **85.90** | **82.88** | 23.7M + 0.4M |
| | FT-KD | 81.50 | 83.50 | **82.95** | 78.50 | 86.15 | 82.52 | 117.7M + 0.4M |
| | FT-SKD | 81.20 | 82.80 | 80.70 | 77.80 | 83.70 | 81.24 | 0.4M + 0.4M |
| | FT-SelKD | **82.75** | **84.20** | 82.85 | **80.35** | **86.35** | **83.30** | 23.7M + 0.4M |
| | AT-KD | 81.05 | 84.35 | 81.60 | 78.10 | 85.05 | 82.03 | 117.7M + 0.4M |
| | AT-SKD | 80.15 | 82.80 | 81.25 | 77.40 | 83.50 | 81.02 | 0.4M + 0.4M |
| | AT-SelKD | **82.45** | **84.70** | **82.20** | **78.50** | **85.30** | **82.63** | 23.7M + 0.4M |
| ResNet-14 | Without KD | 83.60 | 85.40 | 84.10 | 81.20 | 86.55 | 84.17 | 0 + 0.9M |
| | FitNet-KD | 83.60 | 86.00 | 84.85 | 81.55 | 88.00 | 84.80 | 117.7M + 0.9M |
| | FitNet-SKD | 83.00 | 84.05 | 81.80 | 79.10 | 85.00 | 82.59 | 0.9M + 0.9M |
| | FitNet-SelKD | **84.30** | **86.20** | **84.90** | **81.85** | **88.30** | **85.11** | 23.7M + 0.9M |
| | FT-KD | 83.90 | 85.90 | **84.85** | 81.65 | 87.85 | 84.83 | 117.7M + 0.9M |
| | FT-SKD | 83.05 | 85.90 | 82.55 | 79.75 | 86.80 | 83.61 | 0.9M + 0.9M |
| | FT-SelKD | **85.30** | **87.55** | 84.75 | **82.80** | **87.90** | **85.66** | 23.7M + 0.9M |
| | AT-KD | 84.05 | 86.75 | 84.60 | 81.45 | 87.00 | 84.77 | 117.7M + 0.9M |
| | AT-SKD | 83.05 | 83.95 | 82.00 | 79.35 | 85.20 | 82.71 | 0.9M + 0.9M |
| | AT-SelKD | **85.00** | **87.05** | **85.75** | **82.95** | **87.25** | **85.60** | 23.7M + 0.9M |

Then the optimization can be modified as

$$\min_{\theta} CE(\mathbf{Y} \mid \mathbf{P}_s^{\theta}) + \lambda \cdot KL(\mathbf{P}_t \parallel \mathbf{P}_s^{\theta}),$$

$$\text{where} \quad \mathbf{P}_t = \arg \min_{\mathbf{P1} \le \mathbf{1}, \mathbf{1}^{\top}\mathbf{P1}=\gamma} \langle \mathbf{C}_t, \mathbf{P} \rangle - \epsilon H(\mathbf{P}) \tag{14}$$

$$\mathbf{P}_s^{\theta} = \arg \min_{\mathbf{P1} \le \mathbf{1}, \mathbf{1}^{\top}\mathbf{P1}=\gamma} \langle \mathbf{C}_s^{\theta}, \mathbf{P} \rangle - \epsilon H(\mathbf{P})$$

The entire process can be summarized by Algorithm 1. For the prediction in the inference process, we calculate Eq. 14 given the batch testing data. Then for the prediction of sample $i$, we do the $\arg \max$ operation $(\mathbf{P}_s)_{i,j}$ on every $j$ and $1 - \sum_j (\mathbf{P}_s)_{i,j}$ as the result.

## 4 EXPERIMENTS

### 4.1 BASIC SETTINGS

Our experiments are performed using PyTorch 1.4.0 and run on Intel Core i7-7820X CPU @ 3.60GHz with Nvidia GeForce RTX 3080. We take single GPU for classification on CIFAR-10, CIFAR-100 (Krizhevsky et al., 2009) and Tiny ImageNet (Le & Yang, 2015), and evaluate on testing data by top-1 accuracy.

**Experimental Setting Details.** For CIFAR-10 and CIFAR-100, we adopt ResNet50 (He et al., 2016) as the backbone teacher model, while for Tiny ImageNet, we use ResNet32 for training. The settings for students also vary depending on datasets. For CIFAR-10 and CIFAR-100, the experiments of image classification tasks are based on ResNet8 and ResNet14 as the backbone of students. For Tiny ImageNet dataset, we adopt only ResNet8 for training. As for the learning rate, we set 0.05 for all tasks with regard to CIFAR-10 and CIFAR-100 datasets. For Tiny ImageNet, learning rate is 0.2.

As the primary focus of this paper is to pose the problem of selective knowledge distillation, and to find feasible ways to work on the problem, we do not adopt additional specialized techniques to

Table 2: Top-1 accuracy (%) on Tiny ImageNet for closed-set SelKD with ResNet-8 as backbone. We compare the performance of vanilla knowledge distillation (KD) and our selective knowledge distillation (SelKD) in the closed-set SelKD classification tasks.

| Methods | SubTask 0 | SubTask 1 | SubTask 2 | SubTask 3 | SubTask 4 | Mean of SubTasks |
|---|---|---|---|---|---|---|
| Without KD | 52.05 | 53.40 | 53.00 | 50.15 | 54.85 | 52.69 |
| FitNet-KD | 52.55 | 54.00 | 54.10 | 51.25 | 55.60 | 53.50 |
| FitNet-SelKD | **53.20** | **54.95** | **54.80** | **52.05** | **56.40** | **54.28** |
| FT-KD | 52.45 | 53.95 | 54.15 | 52.00 | 55.60 | 53.63 |
| FT-SelKD | **53.15** | **55.05** | **54.70** | **52.10** | **56.55** | **54.31** |
| AT-KD | **52.95** | 54.60 | 54.45 | 51.05 | 55.25 | 53.66 |
| AT-SelKD | 52.90 | **54.70** | **55.20** | **51.90** | **55.90** | **54.12** |

improve performance, such as resampling. This is to control variables and thus all the baselines used in this study represent only the method of knowledge distillation proposed by them.

## 4.2 EXPERIMENTS ON (CLOSED-SET) SELKD

For SelKD, we decompose the overall classification task into various subtasks, with scales varying by dataset. Specifically, for the CIFAR-10 dataset, which consists of 10 classes, we split the task into 2 subtasks, each containing 5 classes. In contrast, CIFAR-100 and Tiny ImageNet, with 100 classes each, are divided into 5 subtasks, each containing 20 classes. We combine feature-based KD methods with different KD frameworks, including vanilla KD, self KD and our SelKD, for a comprehensive comparison to highlight the advantages of our SelKD framework. For feature-based KD, we select typical methods including FitNet (Romero et al., 2014), FT (Kim et al., 2018), and AT (Zagoruyko & Komodakis, 2016). Notably, in both vanilla KD (KD) and self-KD (SKD), we train separate teachers for each subtask, while in SelKD approach, a single teacher is trained to cover all subtasks.

The results of SelKD tasks on the CIFAR-100 and Tiny ImageNet dataset are presented in Table 1 and Table 2, with results for CIFAR-10 is shown in the Appendix A. Our findings clearly indicate that, regardless of the feature-based distillation method employed, SelKD outperforms both KD and SKD in terms of top-1 accuracy. Furthermore, our SelKD method requires fewer parameters compared to KD, which necessitates training an additional teacher for each subtask. Experimental results suggest that a teacher with comprehensive knowledge enhances the performance of subtask students more effectively than multiple teachers with knowledge limited to specific areas. This is because the additional knowledge from the comprehensive teacher aids students in mastering the selected knowledge.

## 4.3 EXPERIMENTS ON OPEN-SET SELKD

We further examine the application of our SelKD method on open-set SelKD tasks. We first apply Algorithm 1 on all the tasks similar to those of closed-set SelKD, but we add an extra class for "not selected" knowledge in each subtask student model. Specifically, for CIFAR-10 dataset, the whole task is separated into 2 subtasks with 6 classes each (5 for selected classes and 1 for all "not selected" classes), while both CIFAR-100 and Tiny ImageNet classification tasks are separated into 5 subtasks with 21 classes each (20 for selected classes and 1 for all "not selected" classes). Table 3 demonstrates the results of open-set SelKD experiments on CIFAR-100.

We have observed that, in line with the results in closed-set SelKD tasks, the experiments conducted on the open-set SelKD tasks reveal that training a group of students with different disjoint tasks is more advantageous when facilitated by a teacher possessing comprehensive knowledge, as opposed to assigning separate teachers who are only well-versed in selected knowledge for each task. This finding also highlights the notable advantage of employing our selective knowledge distillation method in predicting samples that belong to "not selected" knowledge.

Table 3: Top-1 accuracy (%) on CIFAR-100 for open-set SelKD. We compare the performance of vanilla knowledge distillation (KD), self-knowledge distillation (SKD) and our selective knowledge distillation (SelKD) in the open-set SelKD classification tasks. All the comparisons are conducted based on feature-based distillation (FitNet Romero et al. (2014), FT Kim et al. (2018) and AT Zagoruyko & Komodakis (2016)).

| Student Networks | Methods | SubTask 0 | SubTask 1 | SubTask 2 | SubTask 3 | SubTask 4 | Mean of SubTasks | #Param of Teachers + Students |
|---|---|---|---|---|---|---|---|---|
| ResNet-8 | Without KD | 82.19 | 82.82 | 82.62 | 82.44 | 81.42 | 82.30 | 0 + 0.4M |
| | FitNet-KD | 84.65 | 85.20 | 85.69 | 85.72 | 84.20 | 85.10 | 117.7M + 0.4M |
| | FitNet-SKD | 84.77 | 85.41 | 85.60 | 85.54 | 84.62 | 85.19 | 0.4M + 0.4M |
| | FitNet-SelKD | **85.63** | **86.46** | **86.69** | **86.61** | **85.65** | **86.21** | 23.7M + 0.4M |
| | FT-KD | 84.83 | 85.32 | 86.03 | 86.05 | 84.82 | 85.41 | 117.7M + 0.4M |
| | FT-SKD | 84.79 | 85.06 | 85.48 | 85.47 | 84.86 | 85.13 | 0.4M + 0.4M |
| | FT-SelKD | **86.17** | **86.62** | **87.28** | **86.99** | **86.35** | **86.69** | 23.7M + 0.4M |
| | AT-KD | 84.46 | 85.32 | 85.85 | 86.16 | 84.31 | 85.22 | 117.7M + 0.4M |
| | AT-SKD | 84.62 | 85.21 | 85.56 | 85.69 | 84.73 | 85.16 | 0.4M + 0.4M |
| | AT-SelKD | **85.81** | **86.57** | **87.20** | **86.59** | **85.76** | **86.39** | 23.7M + 0.4M |
| ResNet-14 | Without KD | 84.15 | 83.68 | 83.15 | 84.22 | 82.93 | 83.63 | 0 + 0.9M |
| | FitNet-KD | 86.92 | 87.21 | 87.64 | 87.75 | 87.26 | 87.36 | 117.7M + 0.9M |
| | FitNet-SKD | 87.52 | 87.54 | 87.54 | 87.76 | **87.95** | 87.54 | 0.9M + 0.9M |
| | FitNet-SelKD | **87.27** | **87.70** | **87.81** | **87.85** | 87.68 | **87.66** | 23.7M + 0.9M |
| | FT-KD | 87.46 | 88.04 | **88.24** | 88.44 | 88.02 | 88.04 | 117.7M + 0.9M |
| | FT-SKD | 87.33 | 87.99 | 88.04 | 87.96 | 87.92 | 87.86 | 0.9M + 0.9M |
| | FT-SelKD | **87.60** | **88.46** | **88.24** | **88.82** | **88.64** | **88.35** | 23.7M + 0.9M |
| | AT-KD | 86.93 | 87.58 | 87.96 | **88.04** | 87.29 | 87.56 | 117.7M + 0.9M |
| | AT-SKD | 86.54 | 87.13 | 87.15 | 86.61 | 85.90 | 86.67 | 0.9M + 0.9M |
| | AT-SelKD | **87.03** | **87.67** | **88.04** | 88.01 | **87.47** | **87.64** | 23.7M + 0.9M |

Table 4: Ablation Study of SelKD on CIFAR-100.

| Loss Settings | SubTask 0 | SubTask 1 | SubTask 2 | SubTask 3 | SubTask 4 | Mean of SubTasks |
|---|---|---|---|---|---|---|
| Without Distillation | 83.50 | 85.85 | 83.20 | 80.95 | 87.45 | 84.19 |
| Without Logit-based Loss | 83.60 | 86.05 | 83.80 | 80.60 | 85.85 | 83.98 |
| Without Feature-based Loss | 83.95 | 86.15 | 84.70 | 81.80 | 88.15 | 84.95 |
| Ours | **84.30** | **86.20** | **84.90** | **81.85** | **88.30** | **85.11** |

## 4.4 ABLATION STUDY

Regarding the previously presented experiments, the loss function can be divided into three parts, namely from cross-entropy loss with dataset labels, KL divergence with teacher coupling and the divergence with features from the teacher. Here we further explore how the accuracy of the student model is influenced when certain components are absent from the three aforementioned parts. To be more precise, we conduct separate tests to evaluate the impact of each component's absence and compare the obtained results with the original outcome. The results of these tests are summarized and presented in the Table 4.

Based on the analysis of the table data, it is apparent that the removal of any component from the loss function results in a reduction in the accuracy of the student model. Hence, it is imperative to include all parts of losses in the final settings when seeking a more suitable training method for handling Selective KD tasks. Each module plays a crucial role and is indispensable for best performance.

## 5 CONCLUSION

We have proposed a new and practical setting for Knowledge Distillation, called Selective Knowledge Distillation (SelKD), which transfers the partial knowledge to student instead of the whole knowledge in vanilla KD. OT is applied for the SelKD, to help the student learn the subtask. Our current work is focused on classification (including open-set setting).

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

## A  MORE EXPERIMENT RESULT

The result of closed-set SelKD tasks on the CIFAR-10 dataset are presented in Table 5, and it is obvious that SelKD outperforms both KD and SKD in terms of top-1 accuracy.

We further add more experiments for closed-set SelKD on CIFAR-100 with WideResNet-40-2 (Zagoruyko, 2016) as the teacher model and WideResNet-16-2 as the student model. The results are shown in Table 6.

## B  A DETAILED EXPLANATION OF THE BI-LEVEL OPTIMIZATION

We mainly follow (Shi et al., 2023) that understanding or designing the loss via bi-level optimization:

$$\min_{\theta} KL(\mathbf{Y} \mid \mathbf{P}^{\theta}) \quad \text{s.t.} \quad P^{\theta} = \arg\min_{\mathbf{P1=1}} \langle \mathbf{C}^{\theta}, \mathbf{P} \rangle - \epsilon H(\mathbf{P})$$

where $\mathbf{C}^{\theta}$ represents the cosine distance for image feature and text/label feature, with parameters $\theta$, and $\mathbf{Y}$ is the known supervision for learning. As proven in (Shi et al., 2023), $H(\mathbf{P}) = -\langle \mathbf{P}, \log \mathbf{P} - \mathbf{1} \rangle$ is the entropic regularization with coefficient $\epsilon$. The inner optimization is exactly equivalent to the softmax activation, while the outer optimization corresponds to cross-entropy. Thus we can find the above bi-level optimization equals to InfoNCE loss:

$$\min_{\theta} \mathcal{L} = \sum_{i,j} Y_{ij} \log \left( \frac{e^{-C_{ij}^{\theta}/\epsilon}}{\sum_{k} e^{-C_{ik}^{\theta}/\epsilon}} \right)$$

Table 5: Top-1 accuracy (%) on CIFAR-10 for closed-set SelKD. We compare the performance of vanilla knowledge distillation (KD), self-knowledge distillation (SKD) and our selective knowledge distillation (SelKD) in the closed-set SelKD classification tasks.

| Student Networks | Methods | SubTask 0 | SubTask 1 | Mean of SubTasks |
|---|---|---|---|---|
| ResNet-8 | Without KD | 91.28 | 94.76 | 93.02 |
| | FitNet-KD | 91.56 | 95.72 | 93.64 |
| | FitNet-SKD | 91.42 | 95.34 | 93.12 |
| | FitNet-SelKD | **91.88** | **95.88** | **93.88** |
| | FT-KD | 91.98 | 96.22 | 94.10 |
| | FT-SKD | 91.53 | 95.99 | 93.25 |
| | FT-SelKD | **93.68** | **96.50** | **95.09** |
| | AT-KD | 91.66 | 95.60 | 93.63 |
| | AT-SKD | 91.68 | 95.09 | 93.65 |
| | AT-SelKD | **92.90** | **96.12** | **94.51** |
| ResNet-14 | Without KD | 94.42 | 96.54 | 95.48 |
| | FitNet-KD | 94.86 | 96.40 | 95.63 |
| | FitNet-SKD | 91.56 | 95.72 | 93.64 |
| | FitNet-SelKD | **95.04** | **97.72** | **96.38** |
| | FT-KD | 94.56 | 97.04 | 95.80 |
| | FT-SKD | 91.56 | 95.72 | 93.64 |
| | FT-SelKD | **95.66** | **97.92** | **96.79** |
| | AT-KD | 94.02 | 96.76 | 95.39 |
| | AT-SKD | 91.56 | 95.72 | 93.64 |
| | AT-SelKD | **94.84** | **97.64** | **96.24** |

Table 6: Results for closed-set SelKD on CIFAR-100 with WideResNet-40-2 as the teacher model and WideResNet-16-2 as the student model.

| Methods | SubTask 0 | SubTask 1 | SubTask 2 | SubTask 3 | SubTask 4 | Mean |
|---|---|---|---|---|---|---|
| FitNet-KD | 85.67 | 86.52 | 85.14 | 81.97 | 87.72 | 85.40 |
| FitNet-SelKD | **86.15** | **87.68** | **85.83** | **82.76** | **88.04** | **86.09** |
| FT-KD | 85.42 | 85.56 | 86.08 | 82.13 | 88.45 | 85.53 |
| FT-SelKD | **85.92** | **87.23** | **86.17** | **82.56** | **88.73** | **86.12** |
| AT-KD | 85.17 | 85.49 | 85.43 | 81.87 | 87.71 | 85.13 |
| AT-SelKD | **85.55** | **87.25** | **86.38** | **82.41** | **88.52** | **86.02** |

Thus, bi-level optimization is fundamentally a method for designing activation layers or loss functions. In (Shi et al., 2023), modifications to the inner optimization improve the loss. Our work follows this learning framework, but we modify the inner optimization with new constraints to adapt to open-set scenarios, solving it with iterative algorithm (Benamou et al., 2015) to obtain the predicted probability matching matrix. The outer optimization is adjusted to use the original KL Divergence in KD as the loss, resulting in an application in KD problems.

