# OpenReview forum: "SelKD: Selective Knowledge Distillation via Optimal Transport Perspective"
_ICLR.cc/2025/Conference — ICLR 2025 Poster_

### Official Review · Reviewer_bHk7 · 2024-10-22

**Soundness:** 3
**Presentation:** 2
**Contribution:** 3
**Rating:** 6
**Confidence:** 4

**Summary:**

This paper proposes a novel method called Selective Knowledge Distillation (SelKD), which enhances student model performance by transferring only partial knowledge. The integration of Optimal Transport (OT) methods is applied to both closed-set and open-set classification tasks, demonstrating that SelKD outperforms traditional knowledge distillation methods. Overall, the paper offers new insights and practical solutions for the knowledge distillation field.

**Strengths:**

1. The introduction of SelKD allows for effective partial knowledge transfer, reducing redundant learning seen in traditional KD.
2. The method shows high potential for real-world applications, particularly in resource-constrained environments.

**Weaknesses:**

1. The paper lacks concrete details regarding the derivation and experimental implementation. It would be beneficial to provide specific derivations and related code for the experiments to facilitate reproducibility.
2. The analysis of experiments is somewhat superficial, particularly concerning parameter impacts. Given that optimal transport (OT) can be heavily influenced by parameters, providing visual analyses of how these parameters affect the results would enhance understanding.
3. The paper does not include any experiments related to the efficiency of the proposed method. It would be valuable to assess and report the computational efficiency of SelKD in comparison to traditional methods.
4. The experiments primarily focus on ResNet architectures, which restricts the generalizability of the findings. Incorporating more recent distillation methods from 2023 or 2024, as well as testing on different model architectures and datasets, would better demonstrate the effectiveness and versatility of the proposed method. A broader range of experiments would strengthen the paper’s claims regarding SelKD's applicability.

**Questions:**

See above.

---

> ### Author Response · Authors · 2024-11-20
> **Answer**
>
> Thank you for your thorough review. Your comments were particularly helpful to improve our submission. Hopefully, our response below adequately addresses your concerns.
>
> > Q1: The paper lacks concrete details regarding the derivation and experimental implementation.
>
> A1: Sorry for your confusion. For the detailed derivation for open-set SelKD, please refer to Algorithm 1, where $C_s$ and $C_t$ can be regarded as cosine distance of image and label features. And the original algorithm can be found in [1].
>
> For the experimental settings, we train 240 epoch for each model, with learning rate 0.05 for CIFAR-10 and CIFAR-100. In epoch 150, 180, 210, learning rate decays by 90\%. We adopt SGD optimizer, with momentum 0.9. The temperature is 4.0. The core code can be found in this [anonymous link](https://anonymous.4open.science/r/SelKD-431C).
>
> [1] Iterative Bregman Projections for Regularized Transportation Problems.
>
> > Q2: The analysis of experiments is somewhat superficial, particularly concerning parameter impacts. Given that optimal transport (OT) can be heavily influenced by parameters, providing visual analyses of how these parameters affect the results would enhance understanding.
>
> A2: Paper [2], which we have cited in the references, reformulates softmax and cross-entropy into an bi-level optimization with optimal partial transport. The paper points out that the temperature is equivalent to the coefficient of entropic regularization.
>
> Although we introduced OT, it is merely to view the KD problem from a higher-level perspective. Apart from entropic regularization coefficient $\epsilon$, we did not introduce any additional hyperparameters. Moreover, $\epsilon$ is equivalent to the softmax temperature, so we can set this hyperparameter based on other KD-related papers.
>
> [2] Understanding and Generalizing Contrastive Learning from the Inverse Optimal Transport Perspective. ICML, 2023
>
> > Q3: The paper does not include any experiments related to the efficiency of the proposed method. It would be valuable to assess and report the computational efficiency of SelKD in comparison to traditional methods.
>
> A3: We provide the time required to train 5 open-set subtask-specific small models using SelKD and Vanilla KD based on Resnet-50 for 240 epochs each in the table below.
>
> | Methods      | Teacher Training | Student Training | Total Time Cost |
> | ------------ | ---------------- | ---------------- | --------------- |
> | Vanilla KD   | 0.74h\*5         | 1.02h\*5         | 8.80h           |
> | Selective KD | 1.39h            | 1.37h\*5         | 8.24h           |
>
> And as the number of subtasks increases and the performance requirements for the teacher model become higher, the time and resource-saving advantages of SelKD will become more significant.
>
> > Q4: A broader range of experiments would strengthen the paper’s claims regarding SelKD's applicability.
>
> A4: Thank you for your advice, and we further add more experiments with different backbones. The results are shown in the following table. In this setting, we adopt WideResNet-40-2 as the teacher model and WideResNet-16-2 as the student model.
>
>
> | Methods      | SubTask 0 | SubTask 1 | SubTask 2 | SubTask 3 | SubTask 4 | Mean      |
> | ------------ | --------- | --------- | --------- | --------- | --------- | --------- |
> | FitNet-KD    | 85.67     | 86.52     | 85.14     | 81.97     | 87.72     | 85.40     |
> | FitNet-SelKD | **86.15** | **87.68** | **85.83** | **82.76** | **88.04** | **86.09** |
> | FT-KD        | 85.42     | 85.56     | 86.08     | 82.13     | 88.45     | 85.53     |
> | FT-SelKD     | **85.92** | **87.23** | **86.17** | **82.56** | **88.73** | **86.12** |
> | AT-KD        | 85.17     | 85.49     | 85.43     | 81.87     | 87.71     | 85.13     |
> | AT-SelKD     | **85.55** | **87.25** | **86.38** | **82.41** | **88.52** | **86.02** |
>
> For other distillation methods, SelKD is a framework designed to handle new tasks that require transferring subset knowledge. It can be used in conjunction with new distillation methods, as we demonstrate in our experiments, where we provide results combining it with various feature-based distillation methods (FitNet, AT, FT).

---

> ### Author Response · Authors · 2024-11-24
> **Friendly Reminder for Feedback Before Discussion Deadline**
>
> Dear Reviewer,
>
> Thank you very much for your valuable response. As the discussion deadline is approaching, we would be grateful if you could kindly provide your feedback on our response to the concerns you previously raised. We would also be happy to address any further questions or suggestions you may have.
>
> We sincerely appreciate your time and attention.
>
> Best regards,
>
> The Authors

---

> > ### Comment · Reviewer_bHk7 · 2024-11-25
> >
> > For Question 2, I would have preferred to see experiments related to hyperparameter sensitivity. For Question 4, I was hoping to see experiments involving ViT, as the improvement in results for certain architectures is not very significant. I am uncertain whether the observed improvement is due to meticulous hyperparameter tuning. If additional experiments with the same hyperparameters but different random seeds are conducted to observe the stability of the results, I would consider raising my score.

---

> > > ### Author Response · Authors · 2024-11-26
> > > **More Experimental Results**
> > >
> > > The following table shows the results for ViT backbone on CIFAR-100 dataset. The patch size is 4 and the depth is 6 for ViT. We adopt resnet8 as the student model.
> > >
> > > | Methods      | SubTask 0 | SubTask 1 | SubTask 2 | SubTask 3 | SubTask 4 | Mean      |
> > > | ------------ | --------- | --------- | --------- | --------- | --------- | --------- |
> > > | KD    | 80.30     | 82.75   | 83.60     | 80.35     | 84.15     | 82.23     |
> > > | SelKD | **82.90** | **84.00** | **84.35** | **81.95** | **84.40** | **83.52** |
> > >
> > > The following table shows the results for the impact of parameter $\tau$ (temperature of distillation) on the first 2 subtasks. The teacher is resnet101 and student resnet8.
> > >
> > > | $\tau$ | SubTask 0 | SubTask 1 |
> > > | ------ | --------- | --------- |
> > > | 1      | 81.25     | 83.15     |
> > > | 2      | 81.20     | 83.15     |
> > > | 4      | 81.30     | 83.20     |
> > > | 5      | 81.30     | 83.20     |
> > >
> > > We adjusted the random seed and conducted further experiments on the closed-set SelKD of CIFAR-100. The results are shown in the table below. The hyperparameter $\tau$ is fixed to be 4.
> > >
> > > | Random Seed | SubTask 0 | SubTask 1 | SubTask 2 | SubTask 3 | SubTask 4 | Mean  |
> > > | ----------- | --------- | --------- | --------- | --------- | --------- | ----- |
> > > | 0           | 82.95     | 84.45     | 82.25     | 78.85     | 85.90     | 82.88 |
> > > | 42          | 82.48     | 84.33     | 82.72     | 79.02     | 85.44     | 82.80 |
> > > | 100         | 82.79     | 84.87     | 82.24     | 78.89     | 85.88     | 82.93 |
> > >
> > > It is relatively robust to random seeds.

---

> ### Author Response · Authors · 2024-11-26
>
> Dear Reviewer,
>
> Thank you for your insightful feedback and the kindness in raising the score. We will make sure that our discussion and the new findings are reflected in the final version of our paper.
>
> Best regards,
>
> The Authors

---

### Official Review · Reviewer_uP4C · 2024-11-02

**Soundness:** 3
**Presentation:** 2
**Contribution:** 4
**Rating:** 6
**Confidence:** 3

**Summary:**

This paper introduces Selective Knowledge Distillation (SelKD), a novel approach that allows student models to learn only selected subsets of knowledge from teacher models. The authors reformulate knowledge distillation through the lens of Inverse Optimal Transport (IOT) and propose both closed-set and open-set variants of SelKD.

**Strengths:**

**Originality**: The paper offers a fresh perspective on selective knowledge distillation by using optimal transport (OT) to map logit relationships between image and text features. This new approach allows the student model to learn only the relevant subset of knowledge, making it distinct from traditional, full-transfer methods. This new perspective opens up important new research directions.

**Clarity**: The paper is well-structured, with a clear motivation and related work sections that are easy to follow. The optimal transport concepts are complex, but most parts are presented clearly.

**Significance**: Great motivation: enabling selective transfer for specific tasks. The open-set extension further adds value by allowing the model to handle unknown categories, making it versatile for real-world applications.

**Weaknesses:**

**Clarity and Notation**: The notation, especially in the optimal transport section, could be clearer. Key terms and symbols, like
$z$ in Eq. 2 and $\tilde{P}$ in Eq. 7, should be introduced. Additionally, Algorithm 1 may be challenging for readers unfamiliar with classic OT methods. Providing some background or simplifying this section would improve accessibility.

**Computational Complexity**: The paper doesn’t explain the computational cost of the OT-based approach, which may be high given the framework. Information on computation complexity would be helpful to fully understand the method’s practicality and limits.

**Performance**: The impact of random seed on results is not shown. Further testing here could demonstrate robustness and validate improvements.

**Feature-level Transport Potential**: The paper briefly mentions feature-level transport but doesn’t explore it. Expanding this aspect could reveal further benefits and justify using OT for representation learning.

**Questions:**

1. If each subtask requires a separate teacher model, it naturally raises concerns about parameter count and memory costs. Tables presenting parameter counts should clearly explain any dependencies on task splits to avoid misleading impressions.

2. Could you clarify if KL divergence or cross-entropy is minimized in the outer optimization of Eq. 8? The description seems inconsistent.

3. In Figure 2, the green line is shown but not illustrated. Could you clarify this?

4. Switching Tables 2 and 3 order might improve flow.

---

> ### Author Response · Authors · 2024-11-20
> **Answer**
>
> We appreciate your in-depth review and the time you spent providing feedback. We address your concerns as follows:
>
> > Q1: Could you clarify if KL divergence or cross-entropy is minimized in the outer optimization of Eq. 8?
>
> > The paper briefly mentions feature-level transport but doesn’t explore it. Expanding this aspect could reveal further benefits and justify using OT for representation learning.
>
> A1: Thank you for pointing out the shortcomings in our paper. Here, we briefly introduce the main idea we followed, which is based on bi-level optimization to design the loss function.
>
> We mainly follow [1] that understanding or designing the loss via bi-level optimization:
> $$
> \min_\theta KL({Y}\mid P^\theta)\quad
> \text{s.t.} \quad
> {P}^\theta = \arg \min_{P1=1}  \langle  {C}^\theta,{P} \rangle- \epsilon H({P})
> $$
> where ${C}^\theta$ represents the cosine distance for image feature and text/label feature, with parameters $\theta$, and ${Y}$ is the known supervision for learning. As proven in [1], $H({P}) = -\langle{P}, \log {P} - {1}\rangle$ is the entropic regularization with coefficient $\epsilon$. **The inner optimization is exactly equivalent to the softmax activation, while the outer optimization corresponds to cross-entropy.** Thus we can find the above bi-level optimization equals to InfoNCE loss:
>
> $$
> \min_\theta \mathcal{L} = \sum_{i,j}Y_{ij}\log (\frac{e^{-C^\theta_{ij}/\epsilon}}{\sum_{k}e^{-C^\theta_{ik}/\epsilon}})
> $$
>
> Thus, bi-level optimization is fundamentally a method for designing activation layers or loss functions. In [1,2], modifications to the inner optimization improve the loss. **Our work follows this learning framework, but we modify the inner optimization with new constraints to adapt to open-set scenarios, solving it with iterative algorithm [3] to obtain the predicted probability matching matrix.** The outer optimization is adjusted to use the original KL Divergence in KD as the loss, resulting in an application in KD problems.
>
>
> [1] Understanding and generalizing contrastive learning from the inverse optimal transport perspective. ICML, 2023
>
> [2] OT-CLIP: Understanding and generalizing clip via optimal transport. ICML, 2024
>
> [3] Iterative Bregman Projections for Regularized Transportation Problems. SIAM, 2015
>
>
> > Q2: In Figure 2, the green line is shown but not illustrated and switching Tables 2 and 3 order might improve flow.
>
> A2: Sorry for the mistakes, and they should be blue lines. We will re-organize the tables for better flow and fix the mistakes in the updated version.
>
>
> > Q3: The notation, especially in the optimal transport section, could be clearer. Additionally, Algorithm 1 may be challenging for readers unfamiliar with classic OT methods. Providing some background or simplifying this section would improve accessibility.
>
> A3: Thank you for your advice. We will further clarify the algorithm in the updated version. Please refer to [3] Proposition 5 for detailed derivation.
>
> [3] Iterative Bregman Projections for Regularized Transportation Problems. SIAM, 2015
>
> > Q4: Information on computation complexity would be helpful to fully understand the method’s practicality and limits.
>
> A4: We provide the time required to train 5 subtask-specific small models using SelKD and Vanilla KD based on Resnet-50 for 240 epochs each with single GPU in the table below.
>
> | Methods      | Teacher Training | Student Training | Total Time Cost |
> | ------------ | ---------------- | ---------------- | --------------- |
> | Vanilla KD   | 0.74h\*5         | 1.02h\*5         | 8.80h           |
> | Selective KD | 1.39h            | 1.37h\*5         | 8.24h           |
>
> For the computation cost for each student model, SelKD doesn't have advantages. But as the number of subtasks increases and the performance requirements for the teacher model become higher, the time and resource-saving advantages of SelKD will become more significant.
>
> > Q5: The impact of random seed on results is not shown.
>
> A5: We adjusted the random seed and conducted further experiments on the closed-set SelKD of CIFAR-100. The results are shown in the table below.
>
> | Random Seed | SubTask 0 | SubTask 1 | SubTask 2 | SubTask 3 | SubTask 4 | Mean  |
> | ----------- | --------- | --------- | --------- | --------- | --------- | ----- |
> | 0           | 82.95     | 84.45     | 82.25     | 78.85     | 85.90     | 82.88 |
> | 42          | 82.48     | 84.33     | 82.72     | 79.02     | 85.44     | 82.80 |
> | 100         | 82.79     | 84.87     | 82.24     | 78.89     | 85.88     | 82.93 |
>
> It is relatively robust to random seeds.

---

> > ### Comment · Reviewer_uP4C · 2024-11-26
> >
> > Thank you for your response. I appreciate your effort in addressing most of my questions and clarifying the points I raised. Based on your response, I will maintain my positive score.

---

> ### Author Response · Authors · 2024-11-27
>
> Dear Reviewer,
>
> We are delighted that our responses have successfully addressed most of your concerns. Thank you again for your support and the insightful comments you have provided. We are truly grateful for your decision to maintain a positive score. We will make sure that the outcomes of our discussions are integrated into the final version of our paper.
>
> Best regards,
>
> The Authors

---

### Official Review · Reviewer_o62s · 2024-11-08

**Soundness:** 2
**Presentation:** 3
**Contribution:** 3
**Rating:** 6
**Confidence:** 3

**Summary:**

The paper proposes Selective Knowledge Distillation (SelKD) for transferring only selected parts of a teacher model’s knowledge to a student model, framed within the Inverse Optimal Transport (IOT) perspective. This method is relevant for scenarios with resource constraints, where distilling only relevant subsets of knowledge is more practical than full-model distillation. The authors present closed-set and open-set SelKD, where the student learns a subset of categories or can reject classes outside its training scope. Experiments on CIFAR and Tiny ImageNet demonstrate SelKD’s advantages over traditional KD techniques, with improved efficiency and comparable performance on targeted tasks.

**Strengths:**

The paper addresses a practical challenge in knowledge distillation, particularly relevant to deploying models on edge devices with limited resources. By reinterpreting KD within an IOT framework, the authors introduce an innovative approach to selective knowledge transfer that has potential for efficient model adaptation in constrained environments. The experiments on CIFAR and Tiny ImageNet datasets validate the method’s utility, showing that SelKD can effectively focus on specific subtasks with lower computational costs.

**Weaknesses:**

1. The experimental validation is somewhat limited, with results provided only on relatively small datasets (CIFAR and TinyImagenet). The model used in the study is relatively small (ResNet50), with significantly fewer parameters than current large-scale pretrained models. I recommend the authors add experiments using larger backbone models such as ViT.
2. Since SelKD relies on label information, it is challenging to extend it to semi-supervised and unsupervised settings.
3. The open-set SelKD formulation, while promising, lacks guidance on how it could be adapted to real-world scenarios with dynamically changing categories.
4. Lack of theoretical guarantee.

**Questions:**

Would the authors consider exploring SelKD in semi-supervised or unsupervised settings where labeled data for all categories may not be available?

---

> ### Author Response · Authors · 2024-11-20
> **Answer**
>
> Thank you for your detailed review and constructive comments. We hope to thoroughly address all the concerns you highlighted.
>
> > Q1: The experimental validation is limited.
>
> A1: Thank you for your advice, and we further add more experiments with different backbones. The results are shown in the following table. In this setting, we adopt WideResNet-40-2 as the teacher model and WideResNet-16-2 as the student model.
>
>
> | Methods      | SubTask 0 | SubTask 1 | SubTask 2 | SubTask 3 | SubTask 4 | Mean      |
> | ------------ | --------- | --------- | --------- | --------- | --------- | --------- |
> | FitNet-KD    | 85.67     | 86.52     | 85.14     | 81.97     | 87.72     | 85.40     |
> | FitNet-SelKD | **86.15** | **87.68** | **85.83** | **82.76** | **88.04** | **86.09** |
> | FT-KD        | 85.42     | 85.56     | 86.08     | 82.13     | 88.45     | 85.53     |
> | FT-SelKD     | **85.92** | **87.23** | **86.17** | **82.56** | **88.73** | **86.12** |
> | AT-KD        | 85.17     | 85.49     | 85.43     | 81.87     | 87.71     | 85.13     |
> | AT-SelKD     | **85.55** | **87.25** | **86.38** | **82.41** | **88.52** | **86.02** |KD relies on label information, it is challenging to extend it to semi-supervised and unsupervised settings.
>
> > Q2: Since SelKD relies on label information, it is challenging to extend it to semi-supervised and unsupervised settings.
>
> A2: Firstly, If $g(\cdot)$ in Eq. (11) is one-hot function, it degenerates into a label encoder. Therefore, our model can be extended to CLIP-based models.
>
> Secondly, traditional neural networks output logits of size $B\times N$, where $B$ is the batch size and $N$ is the number of classes. In contrast, a CLIP-based model outputs image features of size $B\times k$ and text features of size $N\times k$. By performing the multiplication in Eq. (11), we can obtain logits equivalent to those produced by end-to-end neural networks. Consequently, this ensures that **our model remains compatible with traditional semi-supervised or self-supervised methods typically applied to neural networks**.
>
> Thirdly, compared to end-to-end models, CLIP **has greater potential for self-supervised learning**. Specifically, we can adopt the framework mentioned in the SLIP paper [1], which involves training the teacher model using both contrastive loss for image-text alignment and contrastive loss for image self-supervision.
>
> [1] SLIP: Self-supervision meets Language-Image Pre-training. ECCV, 2022
>
> > Q3: The open-set SelKD formulation, while promising, lacks guidance on how it could be adapted to real-world scenarios with dynamically changing categories.
>
> A3: Take a real-world scenario as an example. Software services are provided to different users and the number of features offered varies depending on their subscription tier. If we regard each type of service as a model, using traditional KD methods would require training a separate teacher model for each user to distill the required functionality. In contrast, with our SelKD approach, a single large model encompassing all functionalities can be used to distill small models tailored to any specific subset of features.
>
> > Q4: Lack of theoretical guarantee.
>
> A4: Thanks for your kind reminder. Indeed, our training process lacks a strong theoretical guarantee, and this is an area we need to explore further in our future work. Our work primarily follows the frameworks proposed in [2,3] for designing the training loss or framework. Regarding the OT algorithm, our theoretical foundation is based on [4], which ensures the convergence of our predictions.
>
> [2] Understanding and Generalizing Contrastive Learning from the Inverse Optimal Transport Perspective. ICML, 2023
>
> [3] OT-CLIP: Understanding and Generalizing CLIP via Optimal Transport. ICML, 2024
>
> [4] Iterative Bregman Projections for Regularized Transportation Problems. SIAM, 2015

---

> > ### Comment · Reviewer_o62s · 2024-11-26
> >
> > Thanks for your rebuttal. It has partially addressed my concerns. After carefully considering the opinion from the other reviewers, I have decided to maintain my positive score.

---

> > > ### Author Response · Authors · 2024-11-26
> > >
> > > Dear Reviewer,
> > >
> > > Thank you very much for your continued support and valuable comments. We appreciate your decision to maintain a positive score, and we are genuinely delighted to learn that our rebuttal has partially addressed your concerns.
> > >
> > > Best regards,
> > >
> > > The Authors

---

### Official Review · Reviewer_9LKC · 2024-11-08

**Soundness:** 2
**Presentation:** 3
**Contribution:** 3
**Rating:** 6
**Confidence:** 4

**Summary:**

This paper discusses the knowledge distillation (KD) problem from the perspective of inverse optimal transport (IOT), introducing a selective KD (SelKD) framework tailored for classification tasks. Specifically, closed-set SelKD allows the student model trained exclusively on the data relevant to specific tasks, while open-set SelKD enables the student model to identify non-selected knowledge to unassigned tasks. Experimental results on image datasets showcase the superior performance of proposed SelKD over the state-of-the-art methods.

**Strengths:**

This paper formulates the KD problem as a bi-level optimization through the perspective of IOT. Building on the IOT-based formulation, this paper further proposes novel closed-set and open-set SelKD methods that allow the student model to learn specific knowledge from assigned tasks, making SelfKD well-suited for real-world, resource-constrained environments.

**Weaknesses:**

However, there are several weaknesses where this paper can improve:
1.  The advantages of the proposed SelKD over vanilla KD are unclear in Figure 1. What are the benefits of training multiple students, each tailored to a specific task, when a general student model showcases consistently strong performance across all tasks in vanilla KD? SelKD could involve higher computational costs due to the need for training multiple task-specific student models.

2. To strengthen the justification for the proposed SelKD in resource-constrained environments, the paper should include additional motivation experiments or theoretical analysis demonstrating that a student model cannot fully receive the knowledge from the teacher model, or in other words, that the teacher fails to transfer essential task-specific knowledge to the student.

3. How are the teacher and student models practically deployed/trained? It is challenging to simultaneously train a cumbersome teacher and student on resource-constrained edge device.

4. In Eqn (11), why is the cost matrix calculated through the multiplication of $f(\cdot)$ and $g(\cdot)$? Is there a specific reason or advantage for using this multiplicative form?

5. The data storage and computational capacity are often limited in resource-constrained environments. Thus, the proposed SelKD is expected to exhibit strong robustness against insufficient training data, while also ensuring efficient computational cost and resource utilization.

6. How are subtasks determined? Can the proposed SelKD consistently maintain superior performance when the categories within the same subtask significantly differ from each other?

7. Some multi-task KD studies [1-3] should be considered as baselines in the experiments.

[1] Raphael Tang, et al. "Distilling task-specific knowledge from BERT into simple neural networks." arXiv, 2019.

[2] Geethu Miriam Jacob, et al. "Online knowledge distillation for multi-task learning." WACV, 2023.

[3] Yangyang Xu, et al. "Multi-task learning with knowledge distillation for dense prediction." ICCV, 2023.

**Questions:**

Please refer to the weaknesses mentioned above.

---

> ### Author Response · Authors · 2024-11-20
> **Answer (1/2)**
>
> Thank you for your time and your critical assessment. Your comments have been very insightful for how our submission can be improved. We hope our responses to your questions below can help to clarify our approach and demonstrate the impact of our work.
>
>
> > Q1: What are the benefits of training multiple students, each tailored to a specific task, when a general student model showcases consistently strong performance across all tasks in vanilla KD?
>
> A1: Knowledge distillation is typically applied to small devices with limited resources and less capable hardware. When the number of classification categories is too large, the performance of small models is significantly worse compared to large models. Small models are more suitable for tasks with fewer categories.
>
> Additionally, in a real-world scenarios where software services are provided to different users, the number of features offered varies depending on their subscription tier. Using traditional KD methods would require training a separate teacher model for each user to distill the required functionality. In contrast, with our SelKD approach, **a single large model encompassing all functionalities can be used to distill small models tailored to any specific subset of features**.
>
> > Q2: SelKD could involve higher computational costs due to the need for training multiple task-specific student models.
>
> A2: We simply divided the tasks to test accuracy, and in our settings, Vanilla KD also requires training student models for each subtask. We provide the time required to train 5 subtask-specific small models using SelKD and Vanilla KD based on Resnet-50 for 240 epochs each in the table below.
>
> | Methods      | Teacher Training | Student Training | Total Time Cost |
> | ------------ | ---------------- | ---------------- | --------------- |
> | Vanilla KD   | 0.74h\*5         | 1.02h\*5         | 8.80h           |
> | Selective KD | 1.39h            | 1.37h\*5         | 8.24h           |
>
> And as the number of subtasks increases and the performance requirements for the teacher model become higher, the time and resource-saving advantages of SelKD will become more significant.
>
> > Q3: To strengthen the justification for the proposed SelKD in resource-constrained environments, the paper should include additional motivation experiments or theoretical analysis demonstrating that a student model cannot fully receive the knowledge from the teacher model.
>
> A3: Large models are typically used on large devices, while small models are often deployed on smaller devices, such as smartphones. The functionality required on smartphones doesn’t need to fully match that of larger devices. Instead, it only needs to meet specific requirements. In our setting, we leverage knowledge distillation to deploy only certain functionalities to smaller devices, enabling the creation of customized models with varying effects and capabilities. Training a dedicated teacher model for each specific functionality would be highly inefficient.
>
> For example, a \\$199 phone requires fewer features (i.e. categories) in its model compared to a \\$599 phone, which demands more functionalities. Instead of training two separate teacher models for the models needed by these two types of devices, training a single large model to perform selective knowledge distillation for both saves resources and time.
>
> > Q4: How are the teacher and student models practically deployed/trained? It is challenging to simultaneously train a cumbersome teacher and student on resource-constrained edge device.
>
> A4: The teacher and student models are both trained on devices with sufficient resources, and the student model is then deployed on resource-constrained edge device.

---

> ### Author Response · Authors · 2024-11-20
> **Answer (2/2)**
>
> > Q5: In Eq. (11), why is the cost matrix calculated through the multiplication of $f(\cdot)$ and $g(\cdot)$? Is there a specific reason or advantage for using this multiplicative form?
>
> A5: For traditional classifiers, $f(\cdot)$ represents the image encoder and $g(\cdot)$ represents the label encoder (e.g., one-hot encoding). For multimodal classifiers based on CLIP, $f(\cdot)$ represents the image encoder while $g(\cdot)$ represents the text encoder. Our formulation generalizes this approach, making it more adaptable to a wider range of methods.
>
> For this multiplicative form, many papers [1,2] adopt it to depict similarity or cost. However, if there is a specific method for cost calculation, it can also be used as a substitute.
>
> [1] Self-supervised Learning of Visual Graph Matching. ECCV, 2022.
>
> [2] Deep Graphical Feature Learning for the Feature Matching Problem. ICCV, 2019.
>
> > Q6: How are subtasks determined? Can the proposed SelKD consistently maintain superior performance when the categories within the same subtask significantly differ from each other?
>
> A6: To better understand the real-world scenario, we simply randomly split the class labels. In fact, for datasets like CIFAR-10 and Tiny ImageNet that are split by class indices, the categories in each subtask are inherently quite distinct from one another.
>
> > Q7: Some multi-task KD studies should be considered as baselines in the experiments.
>
> A7: The goal of multi-task KD [3,4,5] is to enhance the performance of a student model across multiple tasks by transferring knowledge from a teacher model. In contrast, the goal of our SelKD is to transfer a specific portion of the teacher model's knowledge to the student model, thereby optimizing a specific subset of functionalities. The two approaches have distinct goals and are not directly comparable.
>
> [3] Distilling task-specific knowledge from BERT into simple neural networks. arXiv, 2019
>
> [4] Online knowledge distillation for multi-task learning. WACV, 2023
>
> [5] Multi-task learning with knowledge distillation for dense prediction. ICCV, 2023

---

> ### Author Response · Authors · 2024-11-24
> **Friendly Reminder for Feedback Before Discussion Deadline**
>
> Dear Reviewer,
>
> Thank you very much for your valuable response. As the discussion deadline is approaching, we would be grateful if you could kindly provide your feedback on our response to the concerns you previously raised. We would also be happy to address any further questions or suggestions you may have.
>
> We sincerely appreciate your time and attention.
>
> Best regards,
>
> The Authors

---

> ### Comment · Reviewer_9LKC · 2024-11-24
>
> Thank you to the authors for their efforts in addressing my comments. Your responses to both my comments and that of the other reviewers have addressed most of my concerns, and I am pleased to raise my rating score.

---

> > ### Author Response · Authors · 2024-11-25
> >
> > Dear Reviewer,
> >
> > We are very pleased to hear that our rebuttal have adequately addressed your concerns. Thank you for your professional feedback and your kindness in raising the score. We will ensure that our discussion and the additional results are incorporated into the final version of our paper.
> >
> > Best regards,
> >
> > The Authors

---

### Author Response · Authors · 2024-11-23
**General Response**

Dear Area Chairs and Reviewers,

We extend our gratitude to all reviewers for their efforts, insightful comments and constructive suggestions. Overall, the reviewers deem our work as "novel/innovative/fresh" (9LKC, o62s, uP4C), "well-structured" (uP4C), "great motivated" (uP4C), "effective" (o62s, bHk7), "showing potential/well-suited for real-world/resource-constrained environments" (9LKC, o62s, uP4C, bHk7) and "opening up important new research directions" (uP4C).

We make every effort to address reviewers’ concerns and provide additional experimental results. We have updated our revised PDF based on the reviewers' suggestions, with specific changes highlighted in blue. The updates we made are as follows:

1. We conduct more experiments under larger-scaled models. The experimental results on CIFAR-100 dataset demonstrate the effectiveness of our SelKD method under closed-set tasks.
2. We give a clearer explanation on the bi-level optimization and further clarify our OT-based method. The outer optimization is equivalent to cross-entropy, while the inner is equivalent to the softmax activation.
3. We further explain the feature extraction functions $f(\cdot)$ and $g(\cdot)$. In particular, we clarify that the text encoder $g(\cdot)$ degenerates into a label encoder when implemented as a one-hot function.
4. We address formatting issues and fix typos to improve the overall layout and readability of the paper.

In our individual rebuttal, we provide thorough explanations and responses to each of the reviewers' questions. We aim for these responses to effectively address the concerns raised by the reviewers, and we are open to further discussion to ensure a thorough evaluation of our work.

---

### Meta-Review · Area_Chair_BJR5 · 2024-12-08

**Metareview:**

This paper discusses the knowledge distillation (KD) problem from the perspective of inverse optimal transport (IOT), introducing a selective KD (SelKD) framework tailored for classification tasks. The paper addresses a practical challenge in knowledge distillation, particularly relevant to deploying models on edge devices with limited resources. By reinterpreting KD within an IOT framework, the authors introduce an innovative approach to selective knowledge transfer that has potential for efficient model adaptation in constrained environments. The experiments on CIFAR and Tiny ImageNet datasets validate the method’s utility, showing that SelKD can effectively focus on specific subtasks with lower computational costs.

The paper initially has some issues (insufficient experiments and unclear settings). After the rebuttal, I believe most issues have been solved and all reviewers agree to accept.

**Additional Comments On Reviewer Discussion:**

**Summary of Additional Clarifications and Responses:**

- **Expanded Experimental Validation (Q1):**
  - Authors introduced experiments using WideResNet-40-2 as the teacher and WideResNet-16-2 as the student.
  - SelKD demonstrated consistent improvements over standard KD methods across various subtasks.
  - This supports SelKD’s robustness and applicability beyond the initially tested architectures.

- **Applicability to Semi-/Unsupervised Settings (Q2):**
  - By interpreting one-hot labels as a form of label encoding, SelKD can extend to CLIP-based and semi-/self-supervised setups.
  - The authors show that, using CLIP image and text features, they can produce equivalent logits to standard neural networks.
  - Adopting frameworks like SLIP (which employs contrastive and self-supervised losses) can further integrate SelKD into less supervised scenarios.

- **Open-Set SelKD and Real-World Adaptation (Q3):**
  - Authors reaffirm that SelKD’s flexibility allows a single large model to selectively distill subsets of functionalities.
  - This approach is suitable for dynamic, real-world conditions (e.g., different subscription tiers) without the need to retrain separate teacher models for each variant.

- **Theoretical Guarantees (Q4):**
  - While acknowledging the current lack of strong theoretical guarantees, the authors reference related works offering theoretical underpinnings.
  - They highlight this as a future direction for more rigorous theoretical analysis.

---

### Decision · Program_Chairs · 2025-01-22

Accept (Poster)